# Effects of Different Low-Temperature Storage Methods on the Quality and Processing Characteristics of Fresh Beef

**DOI:** 10.3390/foods12040782

**Published:** 2023-02-11

**Authors:** Ruiqi Cao, Lixiu Yan, Shujian Xiao, Bo Hou, Xingchen Zhou, Wei Wang, Ting Bai, Kaixian Zhu, Jie Cheng, Jiamin Zhang

**Affiliations:** 1Meat Processing Key Laboratory of Sichuan Province, College of Food and Biological Engineering, Chengdu University, Chengdu 610106, China; 2Chongqing Academy of Metrology and Quality Inspection, Chongqing 401123, China; 3Cuisine Science Key Laboratory of Sichuan Province, Sichuan Tourism University, Chengdu 610100, China

**Keywords:** fresh beef, low-temperature refrigeration, ice temperature storage, micro-frozen storage, myofibrillar protein

## Abstract

Low-temperature storage has become the most common way for fresh meat storage because of its lower cost and better preservation effect. Traditional low-temperature preservation includes frozen storage and refrigeration storage. The refrigeration storage has a good fresh-keeping effect, but the shelf life is short. Frozen storage has a long shelf life, but it has a great impact on the quality of meat structure and other qualities, and cannot achieve a complete “fresh-keeping” effect. With the development of food processing storage and freezing technology, two new storage methods, ice temperature storage and micro-frozen storage, have attracted more attention. In this paper, the effects of different low-temperature storage methods on the sensory, physicochemical properties, myofibrillar protein oxidation, microstructure, and processing characteristics of fresh beef were studied. The optimal storage methods under different storage requirements were analyzed to reveal the mechanism and efficacy of ice temperature storage and micro-frozen storage technology, as well as the advantages compared with traditional low-temperature refrigeration. It has practical significance for guiding the application of low-temperature storage of fresh meat. Finally, this study concluded that the longest shelf life could be achieved by frozen storage, and the best preservation effect was achieved during the shelf life of ice temperature storage, and the effect of micro-frozen storage on the myofibrillar protein oxidation and microstructure was the best.

## 1. Introduction

China is a major producer and consumer of meat, accounting for 27.19% of the total meat production in the world [1]. As the second largest meat product in China, beef is deeply favored by consumers because of its low fat and high protein.

Meat preservation technology is mainly divided into traditional preservation technology and modern preservation technology. Refrigeration storage and frozen storage are the traditional preservation methods for fresh meat [2], because of its low cost and good preservation effect. However, the shelf life of fresh meat by refrigeration storage is short, while the frozen storage has a long shelf life, but the frozen storage has a great impact on the structure and quality of the meat [3]. The other traditional preservation technology includes vacuum packaging [4], modified atmosphere packaging [5], and food preservatives [6]. Vacuum packaging can effectively extend the shelf life of meat, and its cost is lower than the modified atmosphere [7]. However, vacuum packaging has a poor inhibition of lactic acid bacteria, and has a great impact on the color of the meat [8]. The effect of modified atmosphere packaging is excellent, but the ratio of air conditioning often needs to be determined through experiments [7]. Moreover, the price of equipment required for some modified atmosphere preservation is high, which increases the cost of meat production and processing enterprises. Food preservatives can be divided into natural preservatives and chemical preservatives [9]. Most natural preservatives cannot be mass produced in a short time, and the high cost and poor preservation effect limit the application of natural preservatives in the preservation of fresh meat [10]. At present, chemical preservatives are widely used in meat preservation. Although chemical preservatives have a good effect and low cost, they have a certain toxicity [11]. The massive and long-term consumption of chemical preservatives may seriously affect the health of people, and even increase the risk of cancer [12].

Micro-freezing storage and ice temperature storage are two new low-temperature storage methods [13]. Micro-freezing storage refers to the storage of meat at 1~2 °C below its freezing point, so that the meat is partially frozen but not completely frozen. Qiu et al. [14] compared the effect of refrigeration and micro-freezing on fish fillets, and found that micro-freezing technology could make the lysosome and enzyme activities in micro-frozen fillets always lower than those in refrigeration fillets. Wei et al. [15] took a tilapia fillet stored in micro-frozen storage as the object, and found that micro-frozen storage increased the contents of sweet, salty, and concentrated peptides. Ice temperature storage refers to the storage of meat in the temperature range from 0 °C to the freezing point of meat, so that the meat is not frozen [16]. Ji et al. [17] compared the physicochemical properties of chicken breast muscle during storage at −1.5 °C (ice temperature) and 4 °C (refrigeration). It was found that storage at −1.5 °C could effectively delay the decrease of the solubility, total sulfhydryl and available sulfhydryl contents of chicken breast muscle.

Both micro-frozen storage and ice temperature storage have demonstrated a good effect in the field of aquatic products preservation, but there are few reports in the field of fresh meat preservation [18]. Modern preservation technology includes irradiation technology, coating technology, and ultra-high pressure technology, but these three emerging technologies were less applied in the meat products market in China.

The effects of different low-temperature storage methods on the oxidation and microstructure of fresh meat myofibrillar protein will affect the effect of fresh meat preservation [19]. At present, there are many studies on the effects of refrigeration and freezing on the oxidation and microstructure of fresh myofibrillar protein, but there are few studies on the effects of micro-freezing and ice temperature storage [20].

In this study, fresh beef was selected as the research object. The changes of the physicochemical properties, sensory indexes, myofibrillar protein oxidation, and microstructure of fresh beef during different low-temperature storage methods were studied, and the effects of different low-temperature storage methods on the quality of fresh beef were investigated. This work compares and analyzes the advantages and disadvantages of the traditional refrigeration storage and frozen storage with modern ice temperature storage and micro-frozen storage, revealing the mechanism and key points of micro-frozen storage and ice temperature storage, and provides theoretical support for developing new fresh meat preservation technology.

## 2. Materials and Methods

### 2.1. Materials and Chemicals

The beef of Sichuan scalper hindquarters was provided by Sichuan Zhangfei Beef Co., Ltd (Nanchong, Sichuan, Chian). A principal components analysis (PCA), medium, boric acid, copper sulfate, potassium sulfate, and other reagents were purchased from Chengdu Kelon Chemical Co., Ltd (Chengdu, Sichuan, China). A total sulfhydryl content detection kit and protein carbonyl content detection kit were purchased from Beijing Solebo Technology Co., Ltd (Beijing, China). 2,4,6-Trimethylpyridine was purchased from Sigma-Aldrich Shanghai Co., Ltd (Shanghai, China).

### 2.2. Sample Preparation

Following the removal of fat and surface meat from the fresh beef in an aseptic operation, the beef was divided into meat pieces of about 10 × 5 × 5 cm and weighing 200 g, and then loaded onto trays for packaging. The beef was stored at four low-temperatures: refrigeration, frozen, micro-frozen, and ice temperature. The measurement time of each index was divided into 12 d before storage, once every 2 d, once every 3 d after 12 d and until 18 d.

### 2.3. Physicochemical Properties of the Fresh Meat

#### 2.3.1. Freezing Curve

Beef samples of 5 × 5 × 5 cm were taken to remove visible fat and connective tissue [21]. The metal probe of the temperature recorder was inserted into the center of the beef sample and put into the refrigerator at −18 °C. The temperature of the sample was measured every 10 s to draw the freezing curve of the beef sample.

#### 2.3.2. The Determination of the Chromatic Aberration

The chromatic aberration was determined according to the method of Li et al. [22]. The surface moisture of the sample was wiped dry, and the color difference in the middle of the beef sample was measured with a white board correction meter. The L* value, a* value, and b* value of each group of samples were recorded, and the average value was measured three times in each group.

#### 2.3.3. The Determination of the pH

The pH value of the center of the beef samples in different groups was measured at the depth of 2 cm with the insertion of a pH meter [23]. Each experiment was repeated three times.

#### 2.3.4. The Determination of the Aerobic Count

The plate counting method was used for the determination of the total colony count [24]. The beef sample of 10 g was put into a beaker, and then mixed with 90 mL of 0.85% NaCl solution and shaken thoroughly. The live cells were counted using the plate method after continuous dilution in sterile normal saline. Each experiment was repeated three times.

#### 2.3.5. The Determination of Moisture

About 3 g of complete meat samples were taken from each group and the moisture was measured with a moisture content meter. Each experiment was repeated three times.

#### 2.3.6. The Determination of the Volatile Base Nitrogen

The volatile base nitrogen was determined according to the method of GB5009.228-2016 [25].

#### 2.3.7. The Relative Percentage of Metmyoglobin (MetMb%)

Ten mL of 0.04 M phosphoric acid buffer (pH 6.8) was added to a 10 g beef sample, and then homogenized at high speed at 4 °C for 30 s, placed at 4 °C for 1 h, 10,000 r/min, centrifuged at 10,000 r/min, and 4 °C for 0.5 h. The supernatant of the sample was filtered, and the buffer had a constant volume of 25 mL. The absorbances at 525 nm, 545 nm, 565 nm, and 572 nm were measured. The calculation formula is as follows:$$\mathrm{MetMb}\;\left(\%\right)=(-2.514R_{1}+0.777R_{2}+0.800R_{3}+1.098)\times 100$$
$$R1_{\;}=\frac{A572}{A525},\;R2_{\;}=\frac{A565}{A525},\;R3_{\;}=\frac{A545}{A525}$$

### 2.4. Oxidation and Microstructure of Myofibrillar Protein in Fresh Beef

#### 2.4.1. The Determination of the Total Protein

The total protein content was determined by the Kjeldahl method according to GB5009.5-2016 [26].

#### 2.4.2. The Extraction of the Myofibrillar Protein

The extraction of the myofibrillar protein was in accordance with the method of Wang et al. [27]. The beef sample of 20 g was added to 80 mL buffer solution with a pH 7.0, then homogenized at 10,000× *g* at 4 °C for 1 min, and centrifuged at 10,000× *g* at 4 °C for 15 min. Once the supernatant was discarded the procedure was repeated three times, the precipitation obtained was added to 0.1 mol/L NaCl solution with 4 times the volume. Then, high-speed freezing centrifugation took place at 10,000× *g* for 15 min at 4 °C and repeated three times and filtered using four layers of gauze after the addition of the NaCl solution after the third time. Then, the pH was adjusted to 6.2 with 0.1 mol/L HCl solution under the same conditions, the paste obtained by freezing and centrifugation was the myofibrillar protein.

#### 2.4.3. The Determination of the Total Sulfhydryl Groups

The absorbance at 412 nm of the 0.1 g beef sample was measured using a total sulfhydryl content detection kit [28].

#### 2.4.4. The Determination of the Protein Carbonyl

The 10 g beef sample was processed with a protein carbonyl content detection kit, and its absorbance at 370 nm was measured [29].

#### 2.4.5. The Determination of the Protein Composition

Once the 5% concentrated gel and 12% separated gel were prepared, the electrophoresis tank was installed, and the myofibrillar protein and protein standard extracted from the beef samples were added [30]; after 70 V running for 30 min, 120 V running for 1 h, the electrophoresis gel was dyed and decolorized several times, and then put into the gel imaging system for imaging.

#### 2.4.6. The Determination of the Free Amino Acids

The free amino acids were determined according to the method of GB5009.124-2016 [31]. The 3 g beef sample and 15 mL 6 M hydrochloric acid were placed in the hydrolyzed tube and left to stand for 5 min at −18 °C. The tube was then sealed and heated at 110 °C for 22 h. Then, the solution was cooled to room temperature and diluted to 50 mL, 1 mL was decompressed and dried in a rotary evaporator. Once dried, 1 mL sodium citrate was added to dissolve the solution. It was determined in an amino acid analyzer.

#### 2.4.7. Microstructure Determination

The beef sample was cut with a knife in the direction of the myofibril into cubes with sides about 5 mm long, and it was fixed in glutaraldehyde fixative solution for 24 h. Then, dehydration was repeated with a gradient concentration of ethanol. It was freeze-dried for 12 h and then gold was sprayed, and the results were observed by SEM and photographed.

### 2.5. Processing Characteristics of Fresh Meat

#### 2.5.1. Texture Profile Analysis (TPA)

The beef samples were put into cooking bags and heated in a water bath at 80 °C until the meat center temperature was 75 °C. Following cooling to room temperature, fat and visible connective tissue were removed [32]. The samples were processed into cubes of about 1 cm along the muscle fiber direction and placed on the texture instrument for the TPA measurement, and measured three times for each group and the average value was taken. TPA determination parameters are as follows:

Probe: P36/R; test mode: TPA; pre-test speed: 2.00 mm/SEC, test speed: 2.00 mm/SEC, post-test speed: 5.00 mm/SEC; strain 50.0%, time: 5.00 SEC.

#### 2.5.2. Shear Force Measurement

The samples were processed into cuboids with length, width, and height of about 3 cm, 1 cm, and 1 cm, respectively, along the muscle fiber direction, and placed on the texture instrument for the shear force measurement. Shear force test parameters are as follows:

Probe: HDP/BSW; test mode: compression; pre-test speed: 2.00 mm/SEC, test speed: 1.00 mm/SEC, post-test speed: 10.00 mm/SEC; displacement 25.000 mm, trigger mode: automatic (force), trigger force: 20 g.

#### 2.5.3. The Determination of the Volatile Flavor Substances

The volatile flavor substances were determined according to the method of Chen et al. [33]. Pretreatment conditions: a 3 g beef sample was placed in a 15 mL headspace bottle and sealed at room temperature for 40 min. Then, 1 uL 2,4,6-trimethylpyridine was added.

Quantification: 2,4,6-trimethylpyridine was used as the internal standard for the quantitative analysis, and the absolute content of each substance was calculated according to the following formula:$$\mathrm{Ci}=\frac{\rho \times \nu \times \mathrm{Ai}}{A\times m}$$
where, Ci was the absolute content of each volatile flavor substance/(ug/kg); Ai was the peak area of volatile flavor substances. A was the peak area of the internal standard; M was sample mass/g; ρ was the internal standard mass concentration 2 ug/μL; ν was the internal standard volume/μL.

#### 2.5.4. Based on the PCA Analysis of the Key Flavor Substances

The odor activity value (OAV) was the ratio of the absolute content of each volatile flavor substance to the corresponding threshold value, which could reflect the contribution to the flavor of the fresh beef. When 0 ≤ OAV < 1, it indicated that the substance modified the flavor of the fresh beef; when OAV ≥ 1, it indicated that the substance contributed significantly to the overall flavor of the fresh beef [34].
$$\mathrm{OAV}=\frac{\mathrm{Ci}}{\mathrm{Ti}}$$
where, Ci was the absolute content of each volatile component/(μg/kg); Ti was the sensory threshold of this component/(μg/kg).

## 3. Results and Discussion

### 3.1. Physical and Chemical Properties of the Fresh Beef

#### 3.1.1. The Determination of the Freezing Curve and Freezing Point of Beef

The freezing curve is shown in Figure 1. It can be seen from Figure 1 that the freezing point of beef is −1.6 °C. The temperature of freezing should be set within the range of 0 °C–−1.6 °C, and the micro-freezing temperature should be set between −2.6 °C and −3.6 °C.

#### 3.1.2. Chromatic Aberration

The chromatic aberration values of fresh beef stored at different low-temperatures are shown in Table 1. Color variation is one of the main factors influencing the shelf life of meat. Chromatic aberration, includes the L* value (brightness values), a* value (red), and b* value (yellow). Table 1 shows that the storage times and storage methods have significant effects on the beef off color. In addition to the points set, the three other groups in the L* value are generally declining during the storage process. This is due to the oxidation of myoglobin to metmyoglobin during storage, leading to a dark color and a decreased quality of meat [35]. The L* value of fresh beef was 40.70 after 6 d of refrigeration, and the L* value of the refrigeration group was significantly reduced to 36.72. In the micro-frozen group, the value was 32.68 after 18 d of storage. In both frozen and ice temperature groups, the L* value decreased first and then rose, which was due to the exudation of the beef surface juice, leading to the reflection of light and increasing the brightness value. The L* value was 34.79 after 18 d of frozen storage and 44.09 after 12 d of ice temperature storage.

The positive value of a* indicated the degree of redness. The value of a* in the refrigeration, frozen, and micro-frozen groups showed a decreasing trend, indicating that the redness was decreasing during the storage process, while the ice temperature group showed a rising trend and the redness was deepened. The threshold of the a* value was 14.5. The ice temperature can be stored for a long time to ensure that the beef’s a* value is higher than 14.5. In conclusion, based on the four different low-temperature storage methods, ice temperature storage has a good maintenance of beef color difference value, and micro-freezing can effectively slow down the decline of the beef brightness value at the early stage. Both ice temperature and micro-frozen storage can effectively slow down the decline of color difference value and maintain beef color.

The b* value of fresh beef was 7.88, and the highest b*value in the whole refrigeration stage was 8.22 after 4 d of refrigeration storage, while the positive b* value represented the degree of redness. At 6 d of storage, the b* value of the refrigeration group was 7.91, which was not significantly different from that of the refrigeration group and ice temperature group. The b* value in the micro-frozen group showed a general decreasing trend during the 18 d storage, indicating that the redness was decreasing, which was the same as the change of the a* value in the micro-frozen group.

#### 3.1.3. The pH of the Beef

The pH change of the beef during storage is shown in Figure 2. Figure 2 showed that the pH values in the beef for the four storage methods were on the rise, this was due to the process of microorganisms and enzymes causing the protein to decompose during meat storage, and due to nitrogenous compounds generating alkaline substances. So, with the increase of storage time, the pH of the beef would keep rising.

The pH of fresh beef was 5.50. In the storage process, the pH of the refrigeration group was at a good level on the 2nd day before storage and increased significantly from the 4th day to 5.67 on the 6th day. Compared with refrigeration storage, the ice temperature storage could effectively inhibit the increase of the pH. At 6 d of storage, the pH was 5.57, which was significantly lower than that of the cold storage group (5.67), and at 12 d, the pH reached 5.65. During the whole storage period of 18 d, the pH of beef in the micro-frozen group was better than that in the refrigeration group and the ice temperature group, and the pH of the beef in the micro-frozen group was 5.61 at 12 days, which was significantly lower than that in the ice temperature group. With the increase of storage time, the pH of the frozen beef gradually stabilized during storage, and the pH at 18 d was 5.60, which was significantly lower than 5.68 in the micro-frozen group. The lower-temperature storage may affect the activities of microorganisms and endogenous enzymes, and reduce the decomposition rate of the protein and nitrogenous material, resulting in a slower rise of the pH of the beef.

#### 3.1.4. Aerobic Count

The aerobic count changes of the beef colony under different cryopreservation methods is shown in Figure 3. As could be seen from the Figure 3, both storage time and temperature have a significant impact on the total number of fresh beef colonies. The beef under four different cryogenic preservation methods all exceeded 4 lgCFU/g on the second day and became sub-fresh meat. The colony numbers in beef in refrigeration storage increased the fastest.

The shelf life of beef in frozen storage was about 4–6 d. On the 6th day, the total colony number was 6.97 lgCFU/g, which was significantly higher than 5.17 lgCFU/g in the ice temperature group, 5.23 lgCFU/g in micro-frozen group, and 4.85 lgCFU/g in the frozen group.

The storage life of beef under ice temperature is about 10~12 d. The colony numbers in ice temperature group does not increase significantly with the increase of storage time in the first 10 d of storage. The colony numbers in the freezing temperature group was 6.35 lgCFU/g at 12 d of storage, which was significantly higher than 5.79 lgCFU/g at 10 d of storage, but the time of inhibiting microbial growth and reproduction was longer than that of refrigeration storage. The colony numbers of beef in the micro-frozen group increased significantly with the increase of storage time. The colony numbers of beef in the micro-frozen group was 5.69 lgCFU/g at 10 d of storage, which was significantly lower than that in the ice temperature group, and significantly higher than that in the frozen group. This was because the moisture in the beef in frozen storage was slightly frozen, which had a certain impact on the growth and reproduction of microorganisms.

The shelf life of beef in micro-frozen storage was about 15~18 d. Due to the lowest storage temperature, the colony numbers of beef in frozen storage was still 5.38 lgCFU/g after storage for 18 d, which was less than that of beef in micro-frozen storage after storage for 18 d.

#### 3.1.5. Moisture Content

The variation of moisture content of beef stored at different low-temperatures with storage time is shown in Appendix A. Appendix A shows that the moisture content of beef showed a decreasing trend under different storage methods. With the increase of storage time, the moisture content in beef decreased. Eighty-five percent of moisture in muscle cells was located in myofibril, and the muscle solution after freezing would flow out of the myofibril [36].

The moisture content of the beef in refrigeration storage changed the least and had no significant difference with fresh meat. However, the beef in refrigeration storage would deteriorate after 6 d of storage. During the storage process, there was no significant difference in the moisture content of beef between frozen, micro-frozen, and ice temperature storage. Then, after 18 d, the moisture content of beef in the micro-frozen storage was slightly higher than frozen storage. In conclusion, refrigeration storage had the best moisture retention ability but the storage time was short.

#### 3.1.6. Total Volatile Base Nitrogen (TVB-N)

Figure 4 demonstrates that with the increase of storage time, TVB-N of each storage mode showed an increasing trend. With the help of microorganisms and enzymes, the proteins in beef produced volatile nitrogen and alkaline substances. With the accumulation of these substances, it could affect the odor of the beef, and seriously affect the shelf life of the beef. The TVB-N of fresh beef was 6.55 mg/100 g, and the TVB-N of beef in refrigeration storage was 11.14 mg/100 g after 2 d. The TVB-N with less than 15 mg/100 g, belongs to first-grade freshness. The TVB-N of beef stored at refrigeration for 6 d was 25.6 mg/100 g, which was 25 mg/100 g above the boundary between second-grade freshness and deterioration. The TVB-N of beef stored at ice temperature was 14.49 mg/100 g at 8 d, which was still at first-grade freshness.

Compared with refrigeration storage, the freshness of beef stored at ice temperature could be kept at first-grade freshness for longer, but it was 25.45 mg/100 g at 12 d. The TVB-N of beef in micro-frozen storage at 10 d was 12.39 mg/100 g, which was still at first-grade freshness, The TVB-N of beef in micro-frozen storage at 18 d was 24.29, lower than 25 mg/100 g, which was still at second-grade freshness. The TVB-N of 10 d frozen storage was 12.52 mg/100 g, which was first-grade freshness. Within 15 d of storage, it had no significant difference with the micro-frozen storage group, and on the 18 d, it was 17.23 mg/100 g, which was in the second-grade freshness class, which was significantly lower than that of beef in 18 d micro-frozen storage.

#### 3.1.7. MetMb%

The variation of MetMb% of the beef with different low-temperature storage methods during storage is shown in Figure 5. As can be seen from Figure 5, the MetMb% of beef in the four storage methods showed an increasing trend with the increase of storage time. The MetMb% was related to the freshness of beef, the higher the relative content, the lower the freshness of beef. When MetMb% was lower than 40%, the meat color of beef was better and it was more acceptable to consumers. The MetMb% of fresh beef was the lowest (14.80%), while that of the refrigeration group was over 40% (42.54%) after 6 days of storage. The meat color became worse, which affected the sale of beef. The MetMb% of beef stored in micro-freezing, reached 50.66% and exceeded 40% at 18 d of storage. Under ice temperature storage, beef was still less than 40% (39.41%) after 12 d of storage. The MetMb% of the frozen group was only 31.04% after 18 d of storage, which was better than the other three groups, in terms of the MetMb% of the stored beef.

### 3.2. Oxidation and Microstructure of Myofibrillar Protein in Raw Meat

#### 3.2.1. Changes in the Total Protein Content

The changes in the total protein are shown in Figure 6. The total protein content of the four groups increased with the increase of storage time. The total protein content of fresh beef was 18.685 g/100 g. At 6 d of storage, the total protein content of beef in refrigeration storage was the least, at 28.862 g/100 g. The total protein content was significantly lower than 32.286 g/100 g in the frozen group and 32.816 g/100 g in the micro-frozen group, while there was no significant difference with the beef stored at ice temperature for 6 d of storage. During the whole 18 d of storage, the total protein content of the micro-frozen group was always the highest, but there was no significant difference between the micro-frozen group and the frozen group.

In conclusion, the total protein content of beef under the four low-temperature storage methods showed an increasing trend, and the total protein content of beef in the micro-frozen group was the highest within 18 d of storage, which was not significantly different from that in the frozen group, and the total protein content of beef in the ice temperature storage was also at a high level.

#### 3.2.2. Changes in the Total Sulfhydryl Group Content

The changes of the total sulfhydryl content in beef in different low-temperature storage methods during storage are shown in Figure 7. The total sulfhydryl content of beef under four different storage modes showed a downward trend. The sulfhydryl content of myosin and actin, which constitute myofibrillar, decreased with the increase of storage time in the oxidation reaction, so the change of total sulfhydryl content of beef represented the oxidation degree of protein.

The total sulfhydryl content in fresh beef myofibrillar protein was 4.19 μmol/g. Following 6 d of storage, the total sulfhydryl content in beef in refrigeration storage decreased to 1.81 μmol/g. It was significantly lower than the total sulfhydryl content in the micro-frozen group (3.17 μmol/g), frozen group (3.96 μmol/g), and ice temperature group (2.24 μmol/g). The total sulfhydryl content of the micro-frozen group was 1.67 μmol/g at 12 d, while the total sulfhydryl content of the frozen and ice temperature groups were 2.42 μmol/g and 0.48 μmol/g, respectively. The total sulfhydryl content of beef in frozen storage stored for 18 d was 1.48 μmol/g, which was significantly higher than 0.81 μmol/g of beef in micro-frozen storage.

Compared with the beef in micro-frozen storage stored for 18 d, freezing could better inhibit protein oxidation, because the lower temperature of frozen storage, the higher content of sulfhydryl of myofibrillar protein. Sulfhydryl groups played an important role in the stability of myofibrils, slowing down the formation of disulfide bonds between myosin and actin. The micro-freezing storage temperature was higher than the freezing temperature, the freezing speed was slower, the formation of large ice crystals destroyed the structure of myofibrillar protein, so that the internal sulfhydryl group was exposed, and then oxidized to form a disulfide bond.

#### 3.2.3. Changes in the Protein Carbonyl Content

The Figure 8 shows the changes in the carbonyl content of beef during storage for different storage methods. As could be seen from the Figure 8, the carbonyl content of beef in the four low-temperature storage methods showed an upward trend with the increase of storage time. Therefore, with the increase of storage time, the carbonyl content of beef increased continuously, indicating that the oxidation of beef myofibrillar protein was intensified. As can be seen from Figure 8, the carbonyl content of fresh beef was 3.03 μmol/g. With the increase of storage time, the protein in beef was continuously oxidized and the carbonyl content increased continuously. At 6 d of storage, the carbonyl content of beef in refrigeration storage significantly increased to 16.27 μmol/g. The lowest carbonyl content of beef in frozen storage was 8.80 μmol/g. At 12 d of storage, the carbonyl content of the ice temperature group (23.52 μmol/g) was significantly higher than that of the other two groups, because the refrigeration group had been spoiled. At this time, the carbonyl content of the micro-frozen group was significantly different from that of the beef in frozen storage (18.90 μmol/g and 14.55 μmol/g, respectively). Following 18 d of storage, the carbonyl content of the micro-frozen and frozen groups increased significantly, and the carbonyl content of beef in frozen storage was 21.77 μmol/g, which was significantly lower than that of the micro-frozen group (28.71 μmol/g).

These results indicated that freezing could inhibit the oxidation of beef myofibrillar protein better than micro-freezing and ice temperature after 12 d of storage. While micro-frozen and ice temperature storage were weaker than frozen storage in inhibiting the oxidation of beef myogenic fibrous protein, they were still significantly better than refrigeration storage and had a longer shelf life.

#### 3.2.4. Protein

SDS-PAGE profiles of beef myofibrils from four different cryopreservation methods are shown in Figure 9. The main strip of beef in the map contained 130 kD heavy meromyosin, 70 kD α- actinin, 50 kD troponin, 40 kD actin, and 30 kD actomyosin. [37] Fresh beef strips were darker and clearer. The color of the beef strips after 6 d of refrigeration became lighter, and the changes of other proteins were large except for the small changes of actin at 40 kD, indicating that the protein oxidation was serious. At 6 d of storage, the bands of beef in ice temperature storage were close to those of the beef in frozen storage, but slightly better than those of the beef in refrigeration storage. At 12 d of storage, the bands of beef in ice temperature storage became lighter than those of beef in frozen and micro-frozen storage, and the protein oxidation of beef was more serious than that of the micro-frozen and frozen storage. There was no significant difference in the color between the beef strips of 6 d and 12 d of micro-frozen storage and that of fresh beef, indicating that micro-freezing could better inhibit the oxidation of beef myofibrillar protein. The stripe color of beef in frozen storage was the closest to that of fresh beef during 18 d of storage, and there was no significant difference between the strip color of beef in frozen storage and that of beef in micro-frozen storage during the first 12 d. The strip color of beef in frozen storage at 18 d of storage was darker than that of beef in micro-frozen storage, and the spacing between proteins was clearer, which was better than that of the beef in micro-frozen storage, and the protein degradation degree was the lowest.

#### 3.2.5. Free Amino Acid

Table 2 and Appendix A demonstrate the changes of free amino acid content and relative content of free amino acid in beef in different low-temperature storage methods. Free amino acids are not only important sources of characteristic flavor substances of beef, but are also closely related to the formation of flavor, and are the ultimate products of degradation of macromolecular proteins and peptides. As shown in Table 2, the changes of total amino acids of 16 kinds of free amino acids in beef stored at different low-temperatures showed a trend of first decreasing and then increasing. The amino acid content of beef stored for 6 d in refrigeration storage was 7.47 g/100 g, and the amino acid content of beef stored for 6 d in other ways was better than that stored in refrigeration storage. In addition, the total amino acid content of beef stored in ice temperature was higher than that of beef stored in other low-temperature ways at different storage times.

As can be seen from Appendix A, proline had the highest relative content of amino acids in beef during different low-temperature storage methods. The relative contents of fresh beef: refrigeration 6 d, frozen 6 d, 12 d, 18 d, micro-frozen 6 d, 12 d, 18 d, ice temperature 6 d, 12 d were 11.06%, 11.39%, 11.23%, 10.63%, 9.95%, 10.41%, 11.95%, 10.55%, 10.50%, and 10.35%, respectively. The lowest content was lysine, the relative content was 2.93%, 1.27%, 1.89%, 2.15%, 3.10%, 2.82%, 1.84%, 2.84%, 2.69%, and 2.59%, respectively. In conclusion, taking amino acid as the criterion for judging beef, the ice temperature group was better than that of the micro-frozen group, frozen group, and refrigeration group.

The changes in the absolute content of essential amino acids during beef storage with different low-temperature storage methods are shown in Appendix A. The absolute essential amino acid content of fresh beef was 4.29 g/100 g. With the increase of storage time, the essential amino acid content of the refrigeration group decreased to 3.41 g/100 g at 6 d of storage, which was lower than that of the frozen group (3.85 g/100 g), the micro-frozen group (3.95 g/100 g), and the ice temperature group (4.44 g/100 g). When stored for 12 d, the content of essential amino acids in the ice temperature group was still the highest (4.48 g/100 g), which was higher than that in the frozen group (3.88 g/100 g) and micro-frozen group (3.77 g/100 g). Following 18 d of storage, the contents of essential amino acids in the micro-frozen group was close to those in the frozen group, which were 4.04 g/100 g and 4.02 g/100 g, respectively.

#### 3.2.6. Microstructure

The changes in the microstructure of beef under different low-temperature storage methods are shown in Figure 10. The changes of the beef tissue structure during the storage of different low-temperature storage methods were different. The muscle fibers of fresh samples were arranged neatly and the distance between muscle fibers was small.

With the increase of storage time, the muscle fiber and muscle fiber spacing of beef with four storage methods showed a distinct trend. The space between muscle fibers of beef under refrigeration was larger than that of fresh beef, and muscle fibers were separated. The beef stored at ice temperature also had a large gap, but the structure was more orderly. The muscle fibers of beef stored for 18 d with micro-freezing had a poor structural uniformity and adhesion, and the muscle fiber spacing was larger than that of the beef in frozen storage for 18 d. This might be due to the slow speed of micro-freezing, resulting in a large formation of ice crystals affecting the order of the microstructure of the beef. For the frozen group of beef stored for 18 d, the gap between the beef fibers was still small and this was probably due to the freezing speed. The microstructure damage on the beef was also small and the lower temperature of the frozen beef was generated by the ice crystals. These two reasons may explain why the microscopic structure of the beef in frozen storage was better than that of the beef in micro-frozen storage. In conclusion, the microstructure of the beef was better maintained by refrigeration during short-term storage. The microstructure of beef preserved in frozen storage was better than that in micro-frozen storage and ice temperature storage.

### 3.3. Processing Characteristics of Raw Meat

#### 3.3.1. TPA

As can be seen from Appendix A, the hardness of beef stored in the four storage methods showed a downward trend during storage, and the hardness of fresh beef was the highest, which was 4374 g. With the increase of storage time, the hardness continued to decrease, which may be due to the destruction of the meat structure by microorganisms, leading to the decrease of hardness. In the storage process, the hardness of beef in refrigeration storage decreased significantly only at 2 d of storage, which was 3457 g. In the later storage process, the hardness of beef in refrigeration storage decreased, but did not change significantly. The hardness of beef in the ice temperature group also decreased significantly during storage, and the change of hardness was greater than that of beef in the micro-frozen group and frozen group at the same time. When the beef was stored for 12 d, the hardness of beef in the ice temperature group was 2675 g. The reduction of hardness of beef in the ice temperature group might be due to the effect of microorganisms and endogenous enzymes on the structure of the beef, which reduced its hardness. Micro-freezing and freezing had a good inhibitory effect on the reduction of beef hardness. Following 2 d of storage, the hardness of beef in the micro-frozen group and frozen group was 4183 g and 4660 g, respectively, which were higher than those in the refrigeration group and ice temperature group at the same time. Although the hardness of beef in the micro-frozen group decreased during 2 to 18 d of storage, there was no significant difference. The hardness of the beef in frozen storage was at the highest level during the whole 18 d of storage. The hardness of the beef in frozen storage was 4660 g and 4604 g at 2 d and 4 d, which was significantly higher than that of the fresh beef. Following 6 d of storage, the hardness of the beef in frozen storage was 4354 g. From 8 d to 18 d, the hardness of the frozen group decreased without significant change, and the hardness at 18 d of storage was 2710 g. Although the micro-frozen and frozen storage temperatures were relatively low, which could better inhibit the effects of microorganisms and endoenzymes and reduce their influence on the structure of beef, the ice crystals formed during micro-freezing and freezing could also lead to the change in beef structure, which might be the reason for the decrease of beef hardness in the micro-frozen and frozen groups.

The adhesion of fresh beef was −0.70 g/s, while the adhesion of beef in refrigeration storage decreased with the increase of time during storage. The adhesion of beef in refrigeration storage was −0.32 g/s after 6 d of storage, which still had no significant change with the adhesion of fresh beef. With the increase of time, the adhesion of beef in the ice temperature group gradually decreased after irregular fluctuations, and the value was −0.04 g/s after 12 d of storage. Following 18 d of storage, the values of the micro-frozen group and frozen group were −0.18 g/s and −0.31 g/s, which were significantly lower than those of fresh beef. Elasticity refers to the ability of meat to recover after being extruded by an external force. The elasticity of fresh beef was 0.52, while the elasticity of beef in the refrigeration group had no significant change within 6 d of storage. The elasticity of beef in the refrigeration group fluctuated between 0.51–0.52 in the first 4 days of storage, and the adhesion force was 0.58 after 6 d of storage. There was no significant pattern in the change of beef elasticity with increasing time in other storage methods. The elasticity of the frozen group, the micro-frozen group, and the ice temperature group fluctuated between 0.52–0.65, 0.52–0.73, and 0.47–0.66, respectively. There was no obvious regularity in the return of beef in each group, which fluctuated between 0.17~0.25.

The higher the hardness, the greater the chewiness. The chewiness of beef in each group decreased with the increase of time during storage. The chewiness of fresh beef was 1827 g/s, while the chewiness of beef in refrigeration group decreased significantly during storage, which was 917 g/s at 6 d. The chewiness of the ice temperature group, the micro-frozen group, and frozen group remained corresponding to their hardness, but significantly decreased. For the 12 d of storage in the ice temperature group, and the 18 d of storage for the micro-frozen group and frozen group were 767 g/s, 804 g/s and 767 g/s, respectively.

#### 3.3.2. The Shear Stress

As shown in Appendix A, the beef with the four storage methods showed a downward trend with the increase of storage time. The shear force of the fresh beef was the highest, which was 7916 g. With the increase of storage time, the shear force of the beef in the refrigeration group was as low as 5340 g, which was the lowest level among the four groups. The shear force of the beef in the ice temperature group was 4797 g after 12 d of storage, which was also lower than that of the frozen group and micro-frozen group. When stored for 18 d, the shear forces of the beef in the micro-frozen group and frozen group were 4434 g and 4404 g, respectively.

#### 3.3.3. Volatile Flavor Substances

The changes of volatile flavor substances in beef stored at different low-temperatures are shown in Appendix A. As can be seen from the table, the volatile flavor substances of fresh beef stored at different low-temperatures were determined by SPEM-GC-MS. A total of 32 volatile flavor substances were identified from beef with different storage methods and different storage times, including six aldehydes, two ketones, 10 alcohols, one ester, 12 hydrocarbons, and one acid. Eight volatile flavor substances were identified in fresh beef, including four aldehydes, three alcohols, and one hydrocarbon, with a total value of 614.02 μg/kg. Seventeen volatile flavor substances were detected in fresh beef chilled for 6 d, including four aldehydes, one ketone, seven alcohols, four hydrocarbons, and one acid, with a total value of 1224.00 μg/kg. Nineteen volatile flavor substances were detected in beef in frozen storage on the 6th day, and the most volatile flavor substances were identified in the fresh beef on the sixth day of micro-frozen storage and ice temperature storage, including five aldehydes, two ketones, four alcohols, one ester, and seven hydrocarbons, with a total value of 1575.88 μg/kg. Fourteen volatile flavor substances were detected from the beef in frozen storage on the twelfth day, including five aldehydes, two ketones, three alcohols, and four hydrocarbons, with a total value of 895.27 μg/kg. A total of 12 volatile flavor substances were detected in beef in frozen storage on the eighteenth day, including four aldehydes, two alcohols, and six hydrocarbons, with a total value of 1668.72 μg/kg. A total of 19 volatile flavor substances were detected in beef after 6 d of micro-freezing, including five aldehydes, one ketone, four alcohols, one ester, seven hydrocarbons, and one acid, with a total value of 1897.16 μg/kg. A total of 14 volatile flavor substances were detected in beef after 12 d of micro-freezing, including five aldehydes, one ketone, four alcohols, one ester, and three hydrocarbons, with a total value of 1984.09 μg/kg. A total of 16 volatile flavor substances were detected in beef after micro-frozen storage for 18 d, including four aldehydes, three alcohols, and six hydrocarbons, with a total value of 835.56 μg/kg. A total of 19 volatile flavor substances were detected from beef stored at ice temperature for 6 d, including five aldehydes, one ketone, four alcohols, one ester, and eight hydrocarbons, with a total value of 3239.99 μg/kg. A total of 10 volatile flavor substances were detected from beef stored at ice temperature for 12 d, including four aldehydes, two alcohols, three hydrocarbons, and one acid, with a total value of 1037.67 μg/kg.

#### 3.3.4. PCA

Thirteen key flavor substances (OAV ≥ 1) of beef under different low-temperature storage methods and different storage times were analyzed by PCA analysis. The contribution rates of each principal component to flavor are shown in Appendix A, and the contribution rates of the first four principal components were 38.10%, 18.61%, 13.83%, and 10.51%, respectively. The cumulative contribution rate had reached 81.07%. The PCA analysis can be satisfied by extracting only the principal components containing more than 80% of the information.

The principal component scores of the main flavor substances in beef under different cryogenic storage methods are shown in Figure 11. With the increase of storage time, the flavor substances of beef in the refrigeration group moved from the third quadrant to the second quadrant of the score plot, while the flavor substances of beef in the refrigeration group moved from the third quadrant to the fourth quadrant, and then to the junction of the two and three quadrants, and finally moved back to the fourth quadrant after 18 d of storage. With the increase of storage time, the flavor substances of the beef stored by micro-freezing moved from the third quadrant to the first quadrant, and then moved to the fourth quadrant. Then, the flavor substances of the beef stored on the 18 d moved to the second quadrant. In the principal component score plot, since the position of fresh beef was the farthest from the ice temperature storage for 6 d, it was possible that there were some differences in key flavor substances between fresh beef and beef stored at ice temperature for 6 d. In contrast, beef stored as frozen for 12 d and micro-frozen for 12 d were closer to fresh beef, indicating that in terms of key flavor substances, the key flavor substances of fresh beef were closer to those of beef stored frozen for 12 d and micro-frozen for 12 d.

The loadings of the key flavor substances of beef under different low-temperature storage methods are shown in Figure 12. The absolute value of the loadings of each flavor substance in the loadings indicated the contribution to each principal component, and the larger the absolute value of the flavor substance, the greater the contribution to the principal component. From Figure 12, benzaldehyde, hexanal, 2-heptanone, N-hexyl alcohol, and phenylethyl alcohol, these five flavor substances had a greater effect on the main component 1. 1-octanol, phenylacetaldehyde, 3-methylbutanoic acid, 2-heptanone, and N-hexyl alcohol, these five flavor substances had a greater effect on the main component 2. It was found that there was only fresh beef in the third quadrant and only 1-octanol fell in the third quadrant of the principal component loadings, indicating that 1-octanol was the characteristic flavor substance of fresh beef. The flavor substances of beef stored at ice temperature for 6 d in the first quadrant corresponded to 2-heptanone, N-hexyl alcohol, and 1-octen-3-ol on the flavor substance loadings, indicating that these three substances were the characteristic flavor substances in the flavor substances of beef stored at ice temperature for 6 d.

## 4. Conclusions

This study compares the traditional refrigeration storage and frozen storage with the modern emerging low-temperature storage technology, ice temperature storage, and micro-frozen storage, to study the freshness of different low-temperature storage methods, revealing the mechanism and efficacy of ice temperature storage and micro-freezing technology, as well as the advantages of comparison with traditional storage. Under the experimental conditions, the results of sensory and physicochemical characteristics analysis showed that the shelf life of beef in refrigeration storage was 4–6 d, that of beef in frozen storage was about 12 d, and that of beef in micro-frozen storage was about 18 d. The results of myofibrillar protein oxidation and microstructure showed that the total protein content and total carbonyl content of beef were higher under ice temperature and micro-frozen storage, but the active sulfhydryl content of beef was lower, and the protein degradation degree of beef under ice temperature and micro-frozen storage was the lowest. Compared with the other storage methods, the content of free amino acids in beef under ice temperature storage was the highest and the change of tissue structure was the least, which was better than the other three storage methods.

## Figures and Tables

**Figure 1 foods-12-00782-f001:**
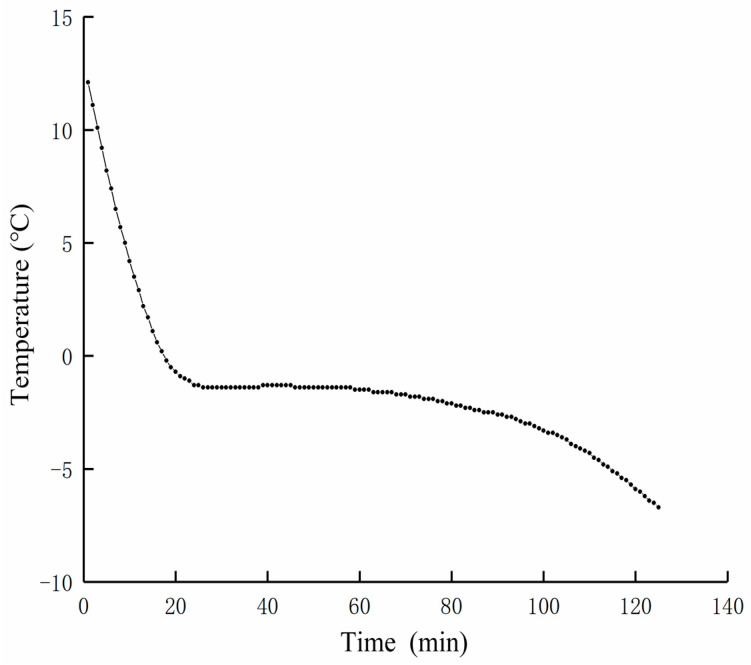
Beef freezing curve.

**Figure 2 foods-12-00782-f002:**
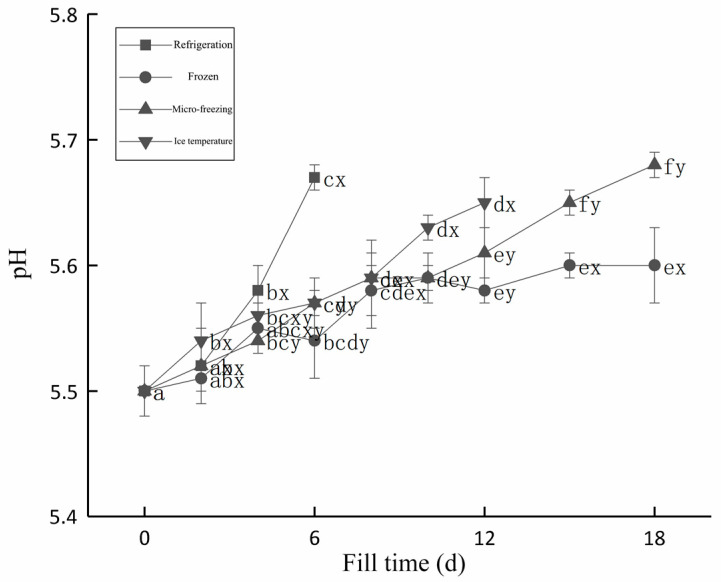
Changes in pH during beef storage with different low-temperature storage methods. d, day.

**Figure 3 foods-12-00782-f003:**
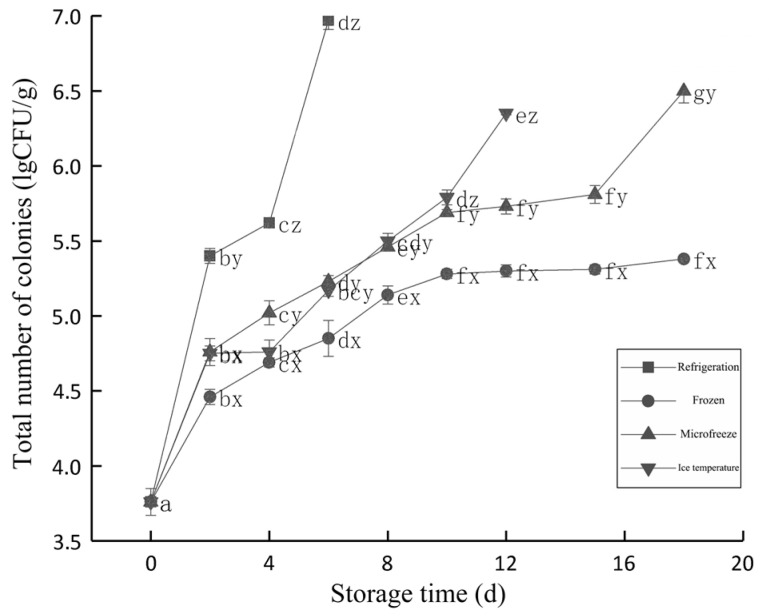
Changes in the total number of colonies in the beef storage with different low-temperature storage methods. d, day.

**Figure 4 foods-12-00782-f004:**
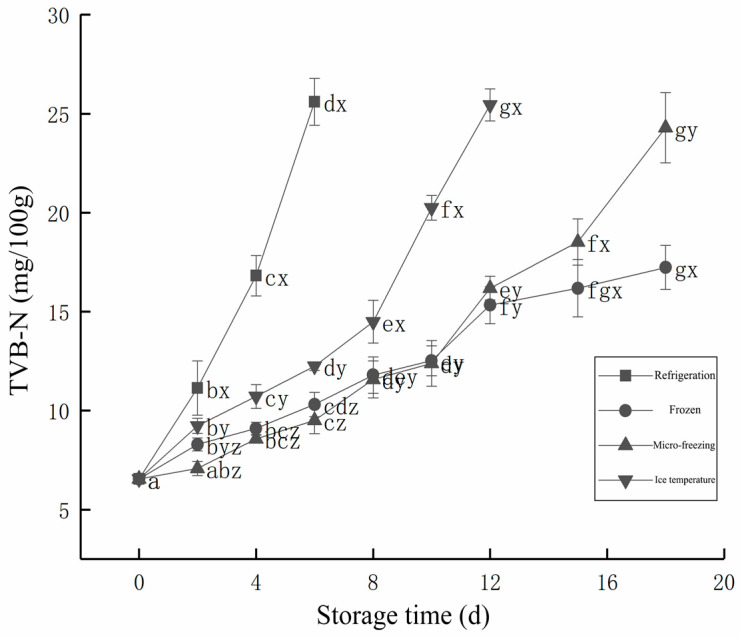
Changes in the TVB-N value of beef in different low-temperature storage methods during storage. d, day.

**Figure 5 foods-12-00782-f005:**
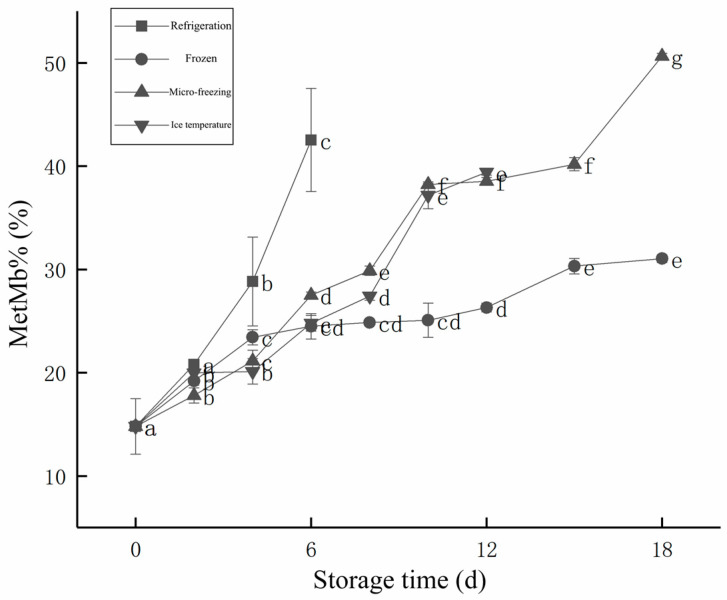
Changes of MetMb% during storage of beef with different cryopreservation methods. d, day.

**Figure 6 foods-12-00782-f006:**
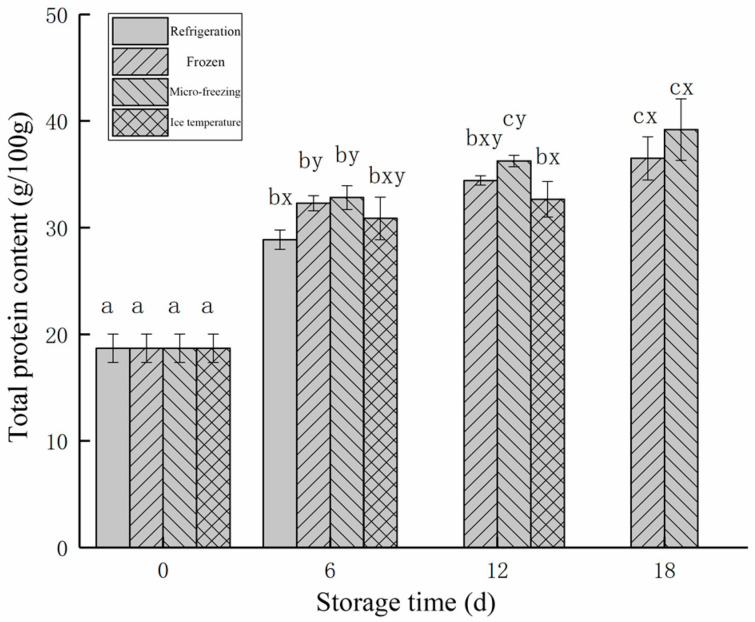
Changes in the total protein content of beef stored in different low-temperature storage methods. d, day.

**Figure 7 foods-12-00782-f007:**
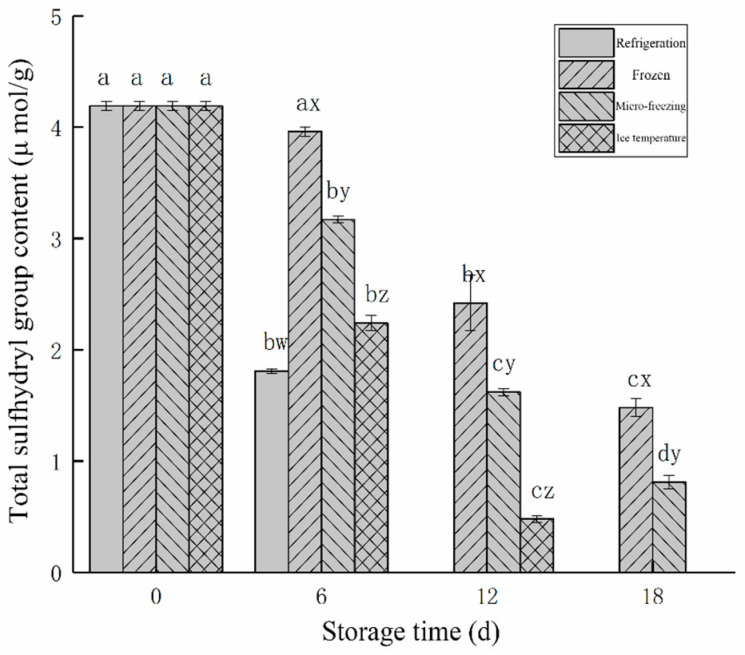
Changes in total sulfhydryl content during beef storage with different low-temperature storage methods. d, day.

**Figure 8 foods-12-00782-f008:**
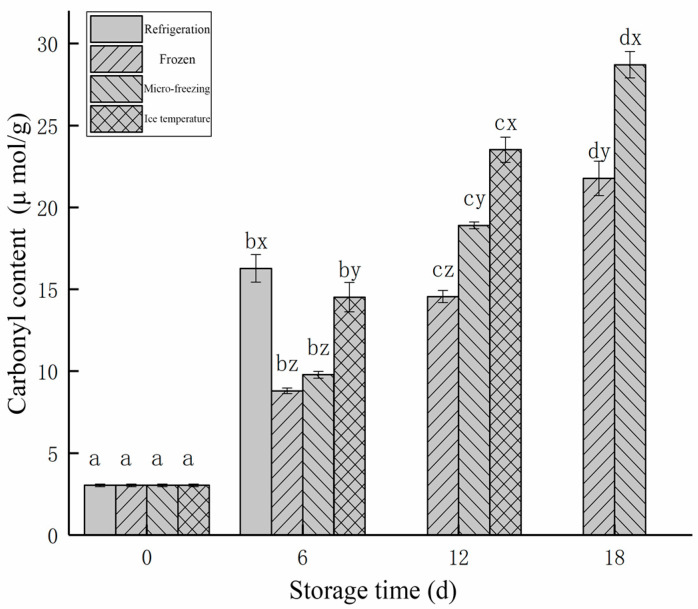
Changes of carbonyl content in beef stored in different low-temperature storage methods. d, day.

**Figure 9 foods-12-00782-f009:**
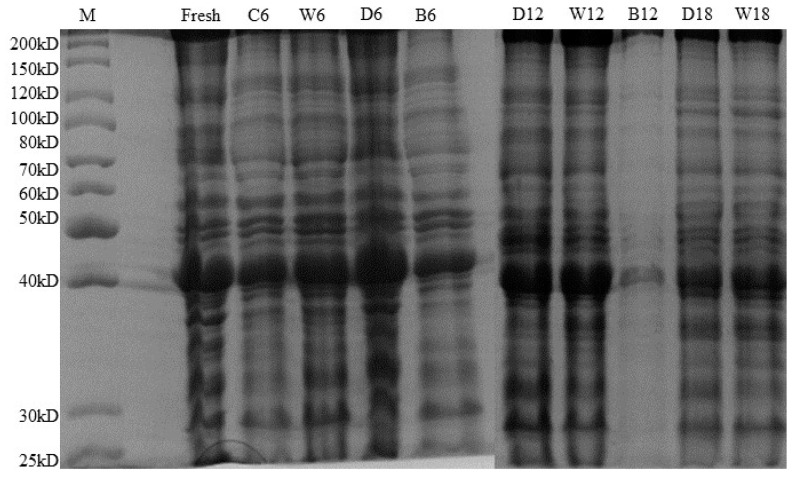
Effect of the SDS-PAGE profile of beef myofibril stored in different low-temperature storage methods. M: marker, fresh: fresh beef, C6: refrigeration 6 d beef, W6: 6 d micro-frozen beef, D6: 6 d frozen beef, B6: 6 d ice temperature beef, D12: 12 d frozen beef, W12: 12 d micro-frozen beef, B12: 12 d ice temperature beef, D18: 18 d frozen beef, W18: 18 d micro-frozen beef.

**Figure 10 foods-12-00782-f010:**
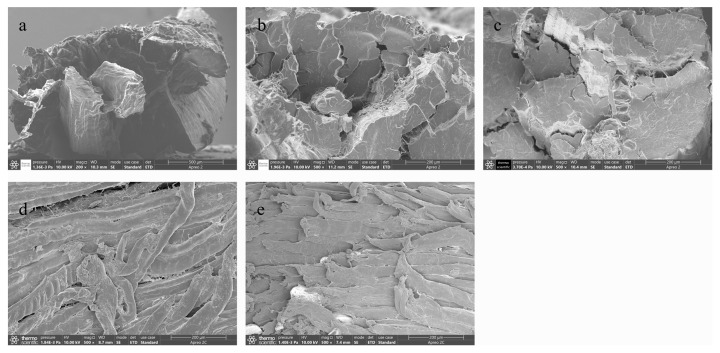
Changes in the beef tissue structure during storage with different low-temperature storage methods. (**a**) fresh beef; (**b**) refrigeration storage for 6 d; (**c**) ice temperature for 12 d; (**d**) micro-frozen for 18 d; (**e**) frozen for 18 d.

**Figure 11 foods-12-00782-f011:**
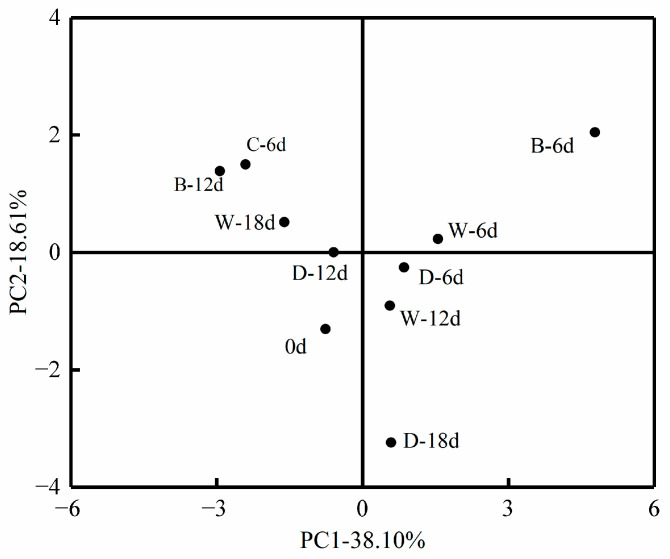
Principal component scores of key flavor substances in beef under different low-temperature storage methods.

**Figure 12 foods-12-00782-f012:**
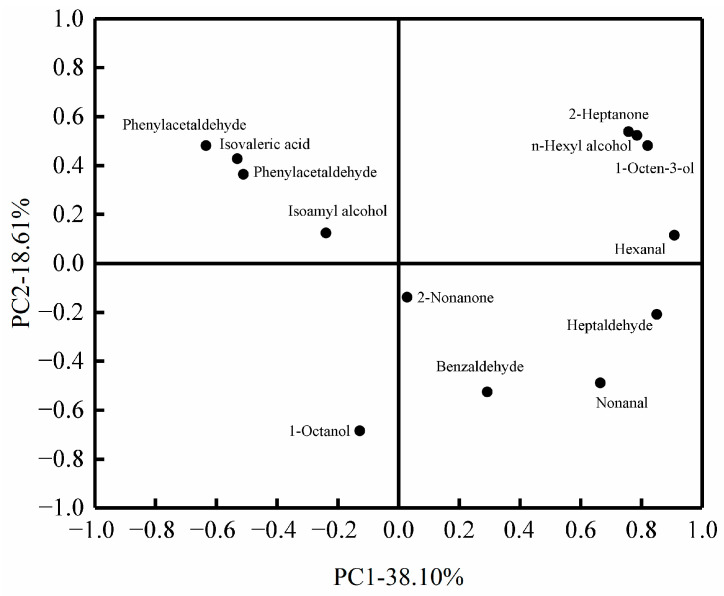
Principal component loading diagram of key flavor compounds in beef under different low-temperature storage methods.

**Table 1 foods-12-00782-t001:** Changes in the color difference value of beef with different low-temperature storage methods.

	Way	Time								
		0 d	2 d	4 d	6 d	8 d	10 d	12 d	15 d	18 d
*L**	Refrigeration	40.70 ± 0.37 ^a^	37.28 ± 1.23 ^bx^	36.84 ± 0.85 ^bxy^	36.72 ± 1.60 ^by^	-	-	-	-	-
	Frozen	40.70 ± 0.37 ^a^	30.31 ± 0.74 ^fy^	32.38 ± 0.14 ^ez^	29.68 ± 1.07 ^fw^	35.34 ± 0.71 ^bcy^	34.08 ± 0.55 ^cdy^	33.31 ± 0.84 ^dey^	36.19 ± 1.49 ^bx^	34.79 ± 0.47 ^bcdx^
	Micro-freezing	40.70 ± 0.37 ^a^	37.42 ± 1.01 ^bx^	35.84 ± 0.47 ^cy^	34.16 ± 1.08 ^dz^	32.91 ± 0.06 ^dez^	32.83 ± 0.67 ^dey^	32.95 ± 0.44 ^dey^	33.83 ± 0.96 ^dex^	32.68 ± 0.16 ^ey^
	Ice temperature	40.70 ± 0.37 ^a^	37.67 ± 1.43 ^cx^	37.75 ± 0.56 ^cx^	43.75 ± 1.15 ^ax^	40.50 ± 1.26 ^bx^	43.62 ± 0.72 ^ax^	44.09 ± 1.23 ^ax^	-	-
*a**	Refrigeration	17.42 ± 0.59 ^a^	13.73 ± 2.05 ^by^	14.39 ± 0.66 ^by^	11.91 ± 1.64 ^by^	-	-	-	-	-
	Frozen	17.42 ± 0.59 ^ab^	18.59 ± 2.35 ^ax^	14.30 ± 0.69 ^cdyz^	13.19 ± 0.70 ^dey^	14.12 ± 0.53 ^cdy^	13.92 ± 0.78 ^cdy^	16.12 ± 1.79 ^bcy^	11.45 ± 0.54 ^ey^	15.07 ± 0.55 ^cdy^
	Micro-freezing	17.42 ± 0.59 ^a^	14.31 ± 1.10 ^by^	13.26 ± 0.42 ^bcz^	13.81 ± 0.71 ^bcy^	11.53 ± 0.90 ^dez^	12.58 ± 1.05 ^cdy^	10.62 ± 0.55 ^ez^	8.84 ± 0.79 ^fx^	7.19 ± 0.61 ^gx^
	Ice temperature	17.42 ± 0.59 ^c^	20.38 ± 1.85 ^bx^	19.72 ± 0.18 ^bcx^	23.82 ± 2.40 ^ax^	19.92 ± 0.59 ^bcx^	22.42 ± 0.91 ^abx^	21.71 ± 1.79 ^abx^	-	-
*b**	Refrigeration	7.88 ± 0.39 ^a^	7.77 ± 1.03 ^ax^	8.22 ± 0.80 ^ax^	7.91 ± 0.60 ^axy^	-	-	-	-	-
	Frozen	7.88 ± 0.39 ^a^	8.31 ± 0.68 ^ax^	5.84 ± 0.26 ^dz^	6.73 ± 0.80 ^bcdy^	6.33 ± 0.31 ^cdy^	4.28 ± 0.11 ^ey^	7.27 ± 1.00 ^abcy^	4.01 ± 0.58 ^ey^	7.20 ± 0.68 ^abcx^
	Micro-freezing	7.88 ± 0.39 ^a^	7.81 ± 0.84 ^ax^	7.18 ± 0.35 ^aby^	6.39 ± 0.67 ^cz^	6.18 ± 0.84 ^by^	7.37 ± 0.47 ^abx^	6.79 ± 1.05 ^abx^	6.72 ± 1.31 ^abx^	6.17 ± 0.30 ^bx^
	Ice temperature	7.88 ± 0.39 ^a^	6.98 ± 1.48 ^abx^	6.53 ± 0.33 ^byz^	8.29 ± 0.36 ^abx^	8.71 ± 0.40 ^ax^	8.30 ± 1.20 ^abx^	6.74 ± 1.37 ^bx^	-	-

Note: ^a–g^ indicates that the indexes of the same storage mode and different storage times have reached significant differences (*p* < 0.05); ^w–z^ indicates that the indexes of different storage methods at the same storage times have reached significant differences (*p* < 0.05). The note in Table 1 is valid for all tables and figures.

**Table 2 foods-12-00782-t002:** Changes of the free amino acid content in beef storage with different low-temperature storage methods (g/100 g).

Amino Acid	Refrigeration		Frozen				Micro-Freezing				Ice Temperature		
	0 d	6 d	0 d	6 d	12 d	18 d	0 d	6 d	12 d	18 d	0 d	6 d	12 d
Asp	0.28	0.22	0.28	0.27	0.23	0.30	0.28	0.27	0.24	0.28	0.28	0.27	0.26
Glu	0.45	0.45	0.45	0.56	0.49	0.62	0.45	0.55	0.51	0.59	0.45	0.59	0.54
Ser	0.35	0.33	0.35	0.36	0.37	0.34	0.35	0.34	0.34	0.34	0.35	0.38	0.40
Gly	0.56	0.42	0.56	0.52	0.49	0.56	0.56	0.48	0.51	0.52	0.56	0.53	0.49
His	0.62	0.43	0.62	0.47	0.53	0.52	0.62	0.45	0.46	0.48	0.62	0.62	0.59
Arg	0.62	0.51	0.62	0.61	0.56	0.56	0.62	0.57	0.51	0.58	0.62	0.61	0.59
Thr	0.70	0.64	0.70	0.71	0.70	0.70	0.70	0.67	0.69	0.70	0.70	0.79	0.76
Ala	0.38	0.26	0.38	0.34	0.32	0.38	0.38	0.33	0.32	0.36	0.38	0.37	0.35
Pro	1.02	0.85	1.02	0.96	0.90	0.87	1.02	0.88	0.99	0.92	1.02	1.00	0.98
Tyr	0.65	0.58	0.65	0.62	0.66	0.58	0.65	0.62	0.64	0.63	0.65	0.71	0.77
Val	0.50	0.39	0.50	0.46	0.44	0.49	0.50	0.49	0.42	0.49	0.50	0.51	0.52
Met	0.47	0.39	0.47	0.39	0.45	0.38	0.47	0.44	0.39	0.40	0.47	0.52	0.57
Ile	0.50	0.41	0.50	0.48	0.45	0.52	0.50	0.48	0.45	0.52	0.50	0.51	0.51
Leu	0.90	0.72	0.90	0.84	0.83	0.91	0.90	0.84	0.83	0.90	0.90	0.91	0.93
Phe	0.95	0.76	0.95	0.81	0.83	0.75	0.95	0.80	0.83	0.78	0.95	0.95	0.95
Lys	0.27	0.09	0.27	0.16	0.18	0.27	0.27	0.24	0.15	0.25	0.27	0.26	0.24
TAA	9.22	7.47	9.22	8.57	8.43	8.75	9.22	8.45	8.30	8.74	9.22	9.51	9.45

## Data Availability

All data generated or analyzed during this study are included in the submitted version of the manuscript.

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
