# Peer review of "Effects of Different Low-Temperature Storage Methods on the Quality and Processing Characteristics of Fresh Beef"

_foods, 2023, doi:10.3390/foods12040782_

Round 1

Reviewer 1 Report

This is an interesting piece of work. However, it needs to be improved.

Principle comments:

1. You have to restrict the number of digit you are quoting to a significant number! It does not make sence to report e.g. 4374.04+- 367.23 ! Sufficient is 4374 +- 367!

2. Many tables are not readable as there are line overflows. Especially Table 1, 3, 5

3. Explanations are missing, especially in Figure 6, 7.

4. You should not quote data in the text which are visible in a table or figure, without need (e.g. for interpretation).

5. In the Introduction you should include detailled information on what has been found on ALL measurements by other authors! You should discuss in the Discussion section how your data relate to what has been found by other authors.

6. It is unclear on what facts you base your conclusion.

7. It is unclear what statistical methods you have applied.

8. You should make clear that the Note in Table 1 is valid for all tables and figures (if it is so??).

9. You should clearly define all abbreviations before using it (e.g. TPA, OAV)

10. All tables and figures which are referenced in the main text of the paper should be included in the paper. Only material which is not necessary to have at hands to understand the content of the paper should be in the Sublementary material. Consequently Figures S.1 to S.5 should be in the paper. Tables (like 3,4 and 5) may be shiften from the paper to the sublementary material.

Author Response

Reviewer #1

This is an interesting piece of work. However, it needs to be improved.

Comment 1:  You have to restrict the number of digit you are quoting to a significant number! It does not make sence to report e.g. 4374.04+- 367.23! Sufficient is 4374 +- 367!

Response:  Thank you for your comments and suggestions. We have revised our manuscript according to your constructive comments.

In lines 419-477 on pages 7-8,"As could be seen from Table 3, the hardness of beef stored in the four storage methods showed a downward trend during storage, and the hardness of fresh beef was the highest, which was 4374.04 g. With the increase of storage time, the hardness continued to decrease, which may be due to the destruction of meat structure by microorganisms, leading to the decrease of hardness. In the storage process, the hardness of beef in refrigeration storage decreased significantly only at 2 d of storage, which was 3457.82 g. In the later storage process, the hardness of beef in refrigeration storage decreased, but didn’t change significantly. The hardness of beef in the ice temperature group also decreased significantly during storage, and the change of hardness was greater than that of beef in the micro-freezing group and frozen group at the same time. When the beef was stored for 12 d, the hardness of beef in the ice temperature group was 2675.05 g. The reduction of hardness of beef in the ice temperature group might be due to the effect of microorganisms and endogenous enzymes on the structure of beef, which reduced its hardness. Micro-freezing and frozen had a good inhibitory effect on the reduction of beef hardness. After 2 d of storage, the hardness of beef in micro-freezing group and frozen group was 4183.53 g and 4660.24 g, respectively, which were higher than those in refrigeration group and ice temperature group at the same time. Although the hardness of beef in micro-freezing group decreased during 2 to 18 d of storage, there was no significant difference. The hardness of the beef in frozen storage was at the highest level during the whole 18 d of storage. The hardness of the beef in frozen storage were 4660.24 g and 4604.8 g at 2 d and 4 d, which were significantly higher than that of the fresh beef. After 6 d of storage, the hardness of the beef in frozen storage was 4354.92 g. From 8 d to 18 d, the hardness of the frozen group decreased without significant change, and the hardness at 18 d of storage was 2710.63 g. Although the micro-freezing and frozen storage temperatures were relatively low, which could better inhibit the effects of microorganisms and endoenzymes and reduce their influence on the structure of beef, the ice crystals formed during micro-freezing and frozen could also lead to the change of beef structure, which might be the reason for the decrease of beef hardness in the micro-freezing and frozen groups.

The adhesion of fresh beef was -0.70 g/s, while the adhesion of beef in refrigeration storage decreased with the increase of time during storage. The adhesion of beef in refrigeration storage was -0.32 g/s after 6 d of storage, which still had no significant change with the adhesion of fresh beef. With the increase of time, the adhesion of beef in the ice temperature group gradually decreased after irregular fluctuations, and the value was -0.04 g/s after 12 d of storage. After 18 d of storage, the values of micro-freezing group and frozen group were -0.18 g/s and -0.31 g/s, which were significantly lower than those of fresh beef. Elasticity refers to the ability of meat to recover after being extruded by external force. The elasticity of fresh beef was 0.52, while the elasticity of beef in refrigeration group had no significant change within 6 d of storage. The elasticity of beef in refrigeration group fluctuates between 0.51-0.52 at the frist 4 days of storage, and the adhesion force was 0.58 after 6 d of storage. There was no significant pattern in the change of beef elasticity with increasing time in other storage methods. The elasticity of frozen group, micro-freezing group and ice temperature group fluctuated between 0.52-0.65, 0.52-0.73 and 0.47-0.66, respectively. There was no obvious regularity in the return of beef in each group, which fluctuated between 0.17~0.25.

The higher the hardness, the greater the chewiness. The chewiness of beef in each group decreased with the increase of time during storage. The chewiness of fresh beef was 1827.28 g.sec, while the chewiness of beef in refrigeration group decreased significantly during storage, which was 917.57 g.sec at 6 d. The chewiness of ice temperature group, micro-freezing group and frozen group remained corresponding to their hardness, but significantly decreased. The 12 d storage in ice temperature group, the 18 d storage in micro-freezing group and frozen group were 767.51 g.sec, 804.99 g.sec and 767.38 g.sec, respectively.

As shown in Table 4, the beef with the four storage methods showed a downward trend with the increase of storage time. The shear force of fresh beef was the highest, which was 7916.42 g. With the increase of storage time, the shear force of beef in refrigeration group was as low as 5340.74 g, which was the lowest level among the four groups. The shear force of beef in ice temperature group was 4797.70 g after 12 d of storage, which was also lower than that of frozen group and micro-freezing group. When stored for 18 d, the shear forces of micro-freezing group and frozen group were 4434.66 g and 4404.80 g, respectively." was revised as “As could be seen from Table S4, the hardness of beef stored in the four storage methods showed a downward trend during storage, and the hardness of fresh beef was the highest, which was 4374 g. With the increase of storage time, the hardness continued to decrease, which may be due to the destruction of meat structure by microorganisms, leading to the decrease of hardness. In the storage process, the hardness of beef in refrigeration storage decreased significantly only at 2 d of storage, which was 3457 g. In the later storage process, the hardness of beef in refrigeration storage decreased, but didn’t change significantly. The hardness of beef in the ice temperature group also decreased significantly during storage, and the change of hardness was greater than that of beef in the micro-freezing group and frozen group at the same time. When the beef was stored for 12 d, the hardness of beef in the ice temperature group was 2675 g. The reduction of hardness of beef in the ice temperature group might be due to the effect of microorganisms and endogenous enzymes on the structure of beef, which reduced its hardness. Micro-freezing and frozen had a good inhibitory effect on the reduction of beef hardness. After 2 d of storage, the hardness of beef in micro-freezing group and frozen group was 4183 g and 4660 g, respectively, which were higher than those in refrigeration group and ice temperature group at the same time. Although the hardness of beef in micro-freezing group decreased during 2 to 18 d of storage, there was no significant difference. The hardness of the beef in frozen storage was at the highest level during the whole 18 d of storage. The hardness of the beef in frozen storage were 4660 g and 4604 g at 2 d and 4 d, which were significantly higher than that of the fresh beef. After 6 d of storage, the hardness of the beef in frozen storage was 4354 g. From 8 d to 18 d, the hardness of the frozen group decreased without significant change, and the hardness at 18 d of storage was 2710 g. Although the micro-freezing and frozen storage temperatures were relatively low, which could better inhibit the effects of microorganisms and endoenzymes and reduce their influence on the structure of beef, the ice crystals formed during micro-freezing and frozen could also lead to the change of beef structure, which might be the reason for the decrease of beef hardness in the micro-freezing and frozen groups.

The adhesion of fresh beef was -0.70 g/s, while the adhesion of beef in refrigeration storage decreased with the increase of time during storage. The adhesion of beef in refrigeration storage was -0.32 g/s after 6 d of storage, which still had no significant change with the adhesion of fresh beef. With the increase of time, the adhesion of beef in the ice temperature group gradually decreased after irregular fluctuations, and the value was -0.04 g/s after 12 d of storage. After 18 d of storage, the values of micro-freezing group and frozen group were -0.18 g/s and -0.31 g/s, which were significantly lower than those of fresh beef. Elasticity refers to the ability of meat to recover after being extruded by external force. The elasticity of fresh beef was 0.52, while the elasticity of beef in refrigeration group had no significant change within 6 d of storage. The elasticity of beef in refrigeration group fluctuates between 0.51-0.52 at the frist 4 days of storage, and the adhesion force was 0.58 after 6 d of storage. There was no significant pattern in the change of beef elasticity with increasing time in other storage methods. The elasticity of frozen group, micro-freezing group and ice temperature group fluctuated between 0.52-0.65, 0.52-0.73 and 0.47-0.66, respectively. There was no obvious regularity in the return of beef in each group, which fluctuated between 0.17~0.25.

The higher the hardness, the greater the chewiness. The chewiness of beef in each group decreased with the increase of time during storage. The chewiness of fresh beef was 1827 g.sec, while the chewiness of beef in refrigeration group decreased significantly during storage, which was 917 g.sec at 6 d. The chewiness of ice temperature group, micro-freezing group and frozen group remained corresponding to their hardness, but significantly decreased. The 12 d storage in ice temperature group, the 18 d storage in micro-freezing group and frozen group were 767 g.sec, 804 g.sec and 767 g.sec, respectively.

As shown in Table S5, the beef with the four storage methods showed a downward trend with the increase of storage time. The shear force of fresh beef was the highest, which was 7916 g. With the increase of storage time, the shear force of beef in refrigeration group was as low as 5340 g, which was the lowest level among the four groups. The shear force of beef in ice temperature group was 4797 g after 12 d of storage, which was also lower than that of frozen group and micro-freezing group. When stored for 18 d, the shear forces of micro-freezing group and frozen group were 4434 g and 4404 g, respectively.”

Table 3. Changes in texture of beef during storage with different low-temperature storage methods. (Before revision)

Time

Way

Hardness(g)

Hardness(g/s)

The elastic

Adhesiveness

Chewiness(g.sec)

Resilience

0 d

Refrigeration

4374.04±367.23c

-0.70±0.14a

0.52±0.01a

0.51±0.06a

1827.28±74.7d

0.20±0.02a

Ice temperature

4374.04±367.23e

-0.70±0.14ab

0.52±0.01a

0.51±0.06a

1827.28±74.7c

0.20±0.02ab

Micro-freezing

4374.04±367.23e

-0.70±0.14abc

0.52±0.01a

0.51±0.05a

1827.28±74.7e

0.20±0.02abc

Frozen

4374.04±367.23c

-0.70±0.14a

0.52±0.01ab

0.51±0.06a

1827.28±74.7c

0.2±0.02abc

2 d

Refrigeration

3457.82±95.69b

-0.59±0.37a

0.51±0.01a

0.58±0.06a

1601.00±220.24c

0.21±0.02a

Ice temperature

3626.00±288.84d

-0.68±0.11abc

0.56±0.13ab

0.62±0.08b

1250.79±319.79b

0.24±0.05bc

Micro-freezing

4183.53±243.46d

-1.04±0.59a

0.7±0.04c

0.53±0.04ab

1730.98±110.65d

0.24±0.02cd

Frozen

4660.24±243.16b

-0.64±0.07ab

0.65±0.01c

0.63±0.01b

1794.83±470.67b

0.24±0.02c

4 d

Refrigeration

3107.01±200.39b

-0.43±0.05a

0.52±0.05a

0.56±0.07a

1130.06±95.65b

0.22±0.01a

Ice temperature

3421.34±115.51cd

-0.78±0.17a

0.58±0.05ab

0.57±0.02ab

1031.58±350.49ab

0.20±0.01ab

Micro-freezing

3653.21±195.84cd

-1.04±0.06a

0.58±0.03ab

0.54±0.02abc

1477.52±108.98c

0.20±0.03ab

Frozen

4604.80±230.8b

-0.63±0.04ab

0.47±0.01a

0.5±0.04a

1551.17±222.45b

0.18±0.02a

6 d

Refrigeration

2656.47±75.56a

-0.32±0.10a

0.58±0.05a

0.63±0.07a

917.57±144.46ab

0.25±0.04a

Ice temperature

3078.91±127.19bc

-0.50±0.08c

0.64±0.01ab

0.61±0.04b

988.77±59..20ab

0.25±0.03c

Micro-freezing

3239.48±366.31bc

-0.81±0.08ab

0.63±0.01b

0.53±0.05abc

1395.61±124.16c

0.24±0.01bcd

Frozen

4354.92±1103.82b

-0.63±0.05ab

0.57±0.05bc

0.62±0.04b

1105.86±70.04a

0.25±0.02c

8 d

Ice temperature

2814.82±329.28b

-0.51±0.03bc

0.59±0.02ab

0.58±0.04ab

985.05±146.09ab

0.18±0.02a

Micro-freezing

3176.12±549.41abc

-0.79±0.17ab

0.62±0.04b

0.56±0.04abcd

1094.73±34.93b

0.24±0.02bcd

Frozen

3409.84±168.58a

-0.51±0.11abc

0.64±0.06c

0.59±0.05ab

966.36±47.33a

0.23±0.04bc

10 d

Ice temperature

2729.67±217.80ab

-0.06±0.02d

0.65±0.04b

0.57±0.03ab

946.66±167.85ab

0.21±0.02abc

Micro-freezing

3154.43±81.93abc

-0.45±0.32bcd

0.73±0.06c

0.57±0.02abcd

918.81±12.17a

0.25±0.03d

Frozen

3386.19±197.6a

-0.47±0.06abc

0.66±0.03c

0.53±0.02a

856.99±29.26a

0.20±0.01ab

12 d

Ice temperature

2675.05±147.77ab

-0.04±0.01d

0.55±0.05ab

0.53±0.03ab

767.51±27.91a

0.17±0.01a

Micro-freezing

2933.70±55.48ab

-0.35±0.12bcd

0.53±0.02a

0.60±0.01bcd

836.35±84.00a

0.21±0.03abc

Frozen

2879.10±173.77a

-0.44±0.23abc

0.57±0.02bc

0.5±0.02a

799.98±165.83a

0.16±0.00a

15 d

Micro-freezing

2901.33±79.22ab

-0.23±0.12cd

0.57±0.02ab

0.61±0.03cd

835.02±47.39a

0.20±0.02ab

Frozen

2739.44±133.57a

-0.38±0.28bc

0.57±0.04bc

0.59±0.09ab

796.97±21.39a

0.23±0.04bc

18 d

Micro-freezing

2630.21±417.71a

-0.18±0.02d

0.62±0.00b

0.63±0.04d

804.99±64.00a

0.19±0.02a

Frozen

2710.63±330.8a

-0.31±0.08c

0.66±0.1c

0.51±0.03a

767.38±96.67a

0.17±0.02a

Table S4. Changes in texture of beef during storage with different low-temperature storage methods. (Revision)

Time

Way

Hardness(g)

Hardness(g/s)

The elastic

Adhesiveness

Chewiness(g.sec)

Resilience

0 d

Refrigeration

4374±367c

-0.70±0.14a

0.52±0.01a

0.51±0.06a

1827±74.7d

0.20±0.02a

Ice temperature

4374±367e

-0.70±0.14ab

0.52±0.01a

0.51±0.06a

1827±74.7c

0.20±0.02ab

Micro-freezing

4374±367e

-0.70±0.14abc

0.52±0.01a

0.51±0.05a

1827±74.7e

0.20±0.02abc

Frozen

4374±367c

-0.70±0.14a

0.52±0.01ab

0.51±0.06a

1827±74.7c

0.2±0.02abc

2 d

Refrigeration

3457±95b

-0.59±0.37a

0.51±0.01a

0.58±0.06a

1601±220c

0.21±0.02a

Ice temperature

3626±288d

-0.68±0.11abc

0.56±0.13ab

0.62±0.08b

1250±319b

0.24±0.05bc

Micro-freezing

4183±243d

-1.04±0.59a

0.7±0.04c

0.53±0.04ab

1730±110d

0.24±0.02cd

Frozen

4660±243b

-0.64±0.07ab

0.65±0.01c

0.63±0.01b

1794±470b

0.24±0.02c

4 d

Refrigeration

3107±200b

-0.43±0.05a

0.52±0.05a

0.56±0.07a

1130±95.6b

0.22±0.01a

Ice temperature

3421±115cd

-0.78±0.17a

0.58±0.05ab

0.57±0.02ab

1031±350ab

0.20±0.01ab

Micro-freezing

3653±195cd

-1.04±0.06a

0.58±0.03ab

0.54±0.02abc

1477±108c

0.20±0.03ab

Frozen

4604±230b

-0.63±0.04ab

0.47±0.01a

0.5±0.04a

1551±222b

0.18±0.02a

6 d

Refrigeration

2656±75a

-0.32±0.10a

0.58±0.05a

0.63±0.07a

917±144ab

0.25±0.04a

Ice temperature

3078±127bc

-0.50±0.08c

0.64±0.01ab

0.61±0.04b

988±59.2ab

0.25±0.03c

Micro-freezing

3239±366bc

-0.81±0.08ab

0.63±0.01b

0.53±0.05abc

1395±124c

0.24±0.01bcd

Frozen

4354±1103b

-0.63±0.05ab

0.57±0.05bc

0.62±0.04b

1105±70a

0.25±0.02c

8 d

Ice temperature

2814±329b

-0.51±0.03bc

0.59±0.02ab

0.58±0.04ab

985±146ab

0.18±0.02a

Micro-freezing

3176±549abc

-0.79±0.17ab

0.62±0.04b

0.56±0.04abcd

1094±34.9b

0.24±0.02bcd

Frozen

3409±168a

-0.51±0.11abc

0.64±0.06c

0.59±0.05ab

966±47.3a

0.23±0.04bc

10 d

Ice temperature

2729±217ab

-0.06±0.02d

0.65±0.04b

0.57±0.03ab

946±167ab

0.21±0.02abc

Micro-freezing

3154±81abc

-0.45±0.32bcd

0.73±0.06c

0.57±0.02abcd

918±12.1a

0.25±0.03d

Frozen

3386±197a

-0.47±0.06abc

0.66±0.03c

0.53±0.02a

856±29.2a

0.20±0.01ab

12 d

Ice temperature

2675±147ab

-0.04±0.01d

0.55±0.05ab

0.53±0.03ab

767±27.9a

0.17±0.01a

Micro-freezing

2933±55ab

-0.35±0.12bcd

0.53±0.02a

0.60±0.01bcd

836±84a

0.21±0.03abc

Frozen

2879±173a

-0.44±0.23abc

0.57±0.02bc

0.5±0.02a

799±165a

0.16±0.00a

15 d

Micro-freezing

2901±79ab

-0.23±0.12cd

0.57±0.02ab

0.61±0.03cd

835±47.3a

0.20±0.02ab

Frozen

2739±133a

-0.38±0.28bc

0.57±0.04bc

0.59±0.09ab

796±21.3a

0.23±0.04bc

18 d

Micro-freezing

2630±417a

-0.18±0.02d

0.62±0.00b

0.63±0.04d

804±64a

0.19±0.02a

Frozen

2710±330a

-0.31±0.08c

0.66±0.1c

0.51±0.03a

767±96.6a

0.17±0.02a

Table 4. Changes of shear force during storage of beef with different low-temperature storage methods. (Before revision)

Time

The shear stress(g)

Refrigeration

Ice temperature

Micro-freezing

Frozen

0 d

7916.42±257.00a

7916.42±257.00a

7916.42±257.00a

7916.42±257a

2 d

7697.07±857.16a

7563.74±136.08a

7487.37±53.46ab

7580.31±236.61ab

4 d

6394.93±559.54b

7287.08±287.62a

7312.40±94.05b

7331.69±33.68ab

6 d

5340.74±255.83b

6430.51±177.96b

6736.49±369.46c

6803.97±1152.25bc

8 d

-

5729.99±541.70bc

6468.80±20.15cd

6280.39±275.93cd

10 d

-

5322.40±723.87cd

6268.08±279.42c

5753.29±367.02d

12 d

-

4797.70±501.64d

5382.16±328.26e

5664.59±115.93de

15 d

-

-

4860.40±55.87f

4779.93±209.22ef

18 d

-

-

4434.66±227.20f

44040.8±308.14f

.

Table S5. Changes of shear force during storage of beef with different low-temperature storage methods. (Revision)

Time

The shear stress(g)

Refrigeration

Ice temperature

Micro-freezing

Frozen

0 d

7916±257a

7916±257a

7916±257a

7916±257a

2 d

7697±857a

7563±136a

7487±53ab

7580±236ab

4 d

6394±559b

7287±287a

7312±94b

7331±33ab

6 d

5340±255b

6430±177b

6736±369c

6803±1152bc

8 d

-

5729±541bc

6468±20cd

6280±275cd

10 d

-

5322±723cd

6268±279c

5753±367d

12 d

-

4797±501d

5382±328e

5664±115de

15 d

-

-

4860±55f

4779±209ef

18 d

-

-

4434±227f

4404±308f

Comment 2:  Many tables are not readable as there are line overflows. Especially Table 1, 3, 5.

Response:  Thank you for your comments and suggestions. We have revised these tables according to your constructive comments.

Comment 3:  Explanations are missing, especially in Figure 6, 7.

Response:  Thank you for your comments and suggestions. Explanations of Fig. 6 and Fig. 7 have been added on pages 9-10 as follows:

The principal component scores of main flavor substances in beef under different cryogenic storage methods are shown in Fig. 11. With the increase of storage time, the flavor substances of beef in the refrigeration group moved from the third quadrant to the second quadrant of the score plot, while the flavor substances of beef in the refrigeration group moved from the third quadrant to the fourth quadrant, and then to the junction of the two and three quadrants, and finally moved back to the fourth quadrant after 18 d of storage. With the increase of storage time, the flavor substances of the beef stored in micro-freezing moved from the third quadrant to the first quadrant, and then moved to the fourth quadrant. After that, the flavor substances of the beef stored on the 18 d moved to the second quadrant. In the principal component score plot, since the position of fresh beef was the farthest from the ice temperature storage for 6 d, it was possible that there were some differences in key flavor substances between fresh beef and beef stored at ice temperature for 6 d. In contrast, beef stored frozen for 12 d and micro-freezing for 12 d were closer to fresh beef, indicating that in terms of key flavor substances, the key flavor substances of fresh beef were closer to those of beef stored frozen for 12 d and micro-freezing for 12 d.

The loadings of the key flavor substances of beef under different low-temperature storage methods are shown in Fig. 12. The absolute value of the loadings of each flavor substance in the loadings indicated the contribution to each principal component, and the larger the absolute value of the flavor substance, the greater the contribution to the principal component. From the Fig. 12, Benzaldehyde, Hexanal, 2-Heptanone, N-Hexyl alcohol and Phenylethyl alcohol, these five flavor substances had a greater effect on the main component 1. 1-Octanol、Phenylacetaldehyde、3-Methylbutanoic acid、2-Heptanone、N-Hexyl alcohol, these five flavor substances had a greater effect on the main component 2. It was found that there was only fresh beef in the third quadrant and only 1-Octanol fell in the third quadrant of the principal component loadings, indicating that 1-Octanol was the characteristic flavor substance of fresh beef. The flavor substances of beef stored at ice temperature for 6 d in the first quadrant corresponded to 2-Heptanone, N-Hexyl alcohol and 1-Octen-3-ol on the flavor substance loadings, indicating that these three substances were the characteristic flavor substances in the flavor substances of beef stored at ice temperature for 6 d.

Comment 4:  You should not quote data in the text which are visible in a table or figure, without need (e.g. for interpretation).

Response:  Thank you for your comments and suggestions. We have revised our manuscript according to your constructive comments.

Comment 5:  In the Introduction you should include detailed information on what has been found on ALL measurements by other authors! You should discuss in the Discussion section how your data relate to what has been found by other authors.

Response:  Thank you for your comments and suggestions. We have revised our manuscript according to your constructive comments.

In line 21-24 on page 3, we have added some detailed information as follows:“Qiu et al. [1] compared the effect of refrigeration and micro-freezing on fish fillets, and found that micro-freezing technology could make the lysosome and enzyme activities in micro-freezing fillets always lower than those in refrigeration fillets. Wei et al. [2] took tilapia fillet stored in micro-freezing storage as the object, and found that micro-freezing stor-age increased the contents of sweet, salty and concentrated peptides.”.

In line 25-27 on page 3, “Ji et al. [3] compared the physicochemical properties of chicken breast muscle during storage at -1.5 °C (ice temperature) and 4 °C (refrigeration).It was found that storage at -1.5 °C could effectively delay the decrease of the solubility, total sulfhydryl and available sulfhydryl contents of chicken breast muscle.”.

  1. Qiu H, Guo X, Deng X, Guo X, Mao X, Xu C, Zhang J: The influence of endogenous cathepsin in different subcellular fractions on the quality deterioration of Northern pike (Esox lucius) fillets during refrigeration and partial freezing storage. Food Science and Biotechnology 2020, 29:-.
  2. Wei P, Zhu K, Cao J, Lin X, Li C: Relationship between Micromolecules and Quality Changes of Tilapia Fillets after Partial Freezing Treatment with Polyphenols. Journal of Agricultural and Food Chemistry 2021.
  3. Ji H, Hou X, Zhang L, Wang X, Chen F: Effect of ice-temperature storage on some properties of salt-soluble proteins and gel from chicken breast muscles. CyTA - Journal of Food 2021, 19:521-531.

Comment 6:  It is unclear on what facts you base your conclusion.

Response: Thank you for your comments and suggestions. We have revised the conclusion according to your constructive comments.

Comment 7:  It is unclear what statistical methods you have applied.

Response: Thank you for your comments and suggestions. In this study, Office Excel 2016, SPSS 25.0 and Origin 2017 were used for statistics, processing, analysis and mapping of experimental data.

Comment 8:  You should make clear that the Note in Table 1 is valid for all tables and figures (if it is so??).

Response: Thank you for your comments and suggestions. We have revised the Note in Table 1 as follows:

Note: a-g indicates that the indexes of the same storage mode and different storage times have reached significant differences (p < 0.05); w-z indicates that the indexes of different storage methods at the same storage time have reached significant differences (p < 0.05). The Note in Table 1 is valid for all tables and figures.

Comment 9:  You should clearly define all abbreviations before using it (e.g. TPA, OAV)

Response: Thank you for your comments and suggestions. We have revised our manuscript according to your constructive comments.

In line 118 on page 6,"TPA" was revised as“Texture Profile Analysis (TPA)”. In line 141 on page 7,"OVA" was revised as“Odor activity value (OAV)”.

Comment 10:  All tables and figures which are referenced in the main text of the paper should be included in the paper. Only material which is not necessary to have at hands to understand the content of the paper should be in the Supplementary material. Consequently Figures S.1 to S.5 should be in the paper. Tables (like 3, 4 and 5) may be shiften from the paper to the supplementary material.

Response: Thank you for your comments and suggestions. Figures S.1-S.5 have been removed into the main text. Table 3-5 have been removed to the Supplementary material, and the serial number of the figures and tables have been modified to:

Figure 1. Beef freezing curve.

Figure 2. Changes in pH during beef storage with different low-temperature storage methods.

Figure 3. Changes in the total number of colonies in beef storage with different low-temperature storage methods

Figure 4. Changes in TVB-N value of beef in different low-temperature storage methods during storage.

Figure 5. Changes of MetMb% during storage of beef with different cryopreservation methods.

Figure 6. Changes of total protein content of beef stored in different low-temperature storage methods.

Figure 7. Changes in total sulfhydryl content during beef storage with different low-temperature storage methods.

Figure 8. Changes of carbonyl content in beef stored in different low-temperature storage methods.

Figure 9. Effect of SDS-PAGE profile of beef myofibril stored in different low-temperature storage methods.

Figure 10. Changes in beef tissue structure during storage with different low-temperature storage methods.

Figure 11. Principal component scores of key flavor substances in beef under different low-temperature storage methods.

Figure 12. Principal component loading diagram of key flavor compounds in beef under different low-temperature storage methods.

Table S1: Changes in moisture content of raw beef under different low-temperature storage methods.

Table S2: Changes of relative content of free amino acids in beef storage with different low-temperature storage methods (%).

Table S3: Changes in absolute content of essential amino acids during storage of beef with different low-temperature storage methods (g/100g).

Table S4: Changes in texture of beef during storage with different low-temperature storage methods.

Table S5: Changes of shear force during storage of beef with different low-temperature storage methods.

Table S6: Changes of Volatile Flavor Substances of Beef in Different Low-temperature Storage.

Table S7: OAV of volatile flavor compounds in beef under different low-temperature storage methods.

Table S8: Principal component eigenvalues and cumulative contribution rate.

Reviewer 2 Report

Needs extensive revision of English language usage. Tables 1 and 5 need to be reformatted.

Please see my suggestions on the attached copy

Author Response

Reviewer #2

Needs extensive revision of English language usage. Tables 1 and 5 need to be reformatted. Please see my suggestions on the attached copy.

Response: Thank you for your affirmation and comments. We have revised our manuscript according to your comments. Please see the attachment.

Reviewer 3 Report

This research deals with a study that compares and analyzes the advantages and disadvantages of traditional refrigeration storage and frozen storage with modern ice temperature storage and micro-freezing storage reveals the mechanism and key points of micro-freezing storage and ice temperature storage and provides theoretical support for developing new fresh meat preservation technology.

The topic is within the scope of the Journal, the topic is interesting, the manuscript is well-written, and the analyzed parameters are interesting. However, minor revisions are requested. Please see the following comments.

Abstract

Line 2. What do you mean by "lower costs"?

Line 2. Please replace the word refrigeration here, "Traditional low-temperature refrigeration", with "preservation".

Line 5. Please consider using the plural of the word "quality" here.

Lines 7-8. Please consider rewriting this sentence "With the progress of food processing, refrigeration storage and freezing technology, the cold preservation technology between traditional refrigeration storage and frozen storage have attracted more attention, which is ice temperature storage and micro-freezing storage." for a better understanding.

The abstract lacks results (please summarize the article's main findings) and conclusions (please indicate the main conclusions or interpretations).

Introduction

Line 7. Is it low cost?

Please delete references from the end of the following since these are the results of this study "In this study, fresh beef was selected as the research object. The changes of physicochemical properties, sensory indexes, myofibrillar protein oxidation and microstructure  of fresh beef during different low-temperature storage method were studied, and the effects of different low-temperature storage method on the quality of fresh beef were investigated. This work compares and analyzes the advantages and disadvantages of the traditional refrigeration storage and frozen storage with modern ice temperature storage and micro-freezing storage, reveals the mechanism and key points of micro-freezing storage and ice temperature storage, and provides theoretical support for developing new fresh meat preservation technology [18, 19].".

Materials and Methods

2.1. Sample preparation. Principal components analysis (PCA) medium?

Please reformulate this sentence "The measurement time of each index was divided into 12 d before storage, once every 2 d, once every 3 d after 12 d until 18 d." for a better understanding.

Author Response

Reviewer #3

This research deals with a study that compares and analyzes the advantages and disadvantages of traditional refrigeration storage and frozen storage with modern ice temperature storage and micro-freezing storage reveals the mechanism and key points of micro-freezing storage and ice temperature storage and provides theoretical support for developing new fresh meat preservation technology.

The topic is within the scope of the Journal, the topic is interesting, the manuscript is well written, and the analyzed parameters are interesting. However, minor revisions are requested.

Abstract

Comment 1: Line 2. What do you mean by "lower costs"?

Response: Thank you for your affirmation and comments. The cost of low-temperature storage is lower than other storage methods. Refrigeration storage and frozen storage can usually be done in a refrigerator or freezer, or even in a naturally low temperature environment, whereas other novel storage methods, such as the ice temperature and micro-freezing methods used in this article, require other expensive equipment.

Comment 2: Line 2. Please replace the word refrigeration here, "Traditional low-temperature refrigeration", with "preservation".

Response: Thank you for your comments and suggestions. We have revised our manuscript according to your constructive comments.

In the line 2 of the abstract, "Traditional low-temperature refrigeration includes frozen storage and refrigeration storage." was revised as“Traditional low-temperature preservation includes frozen storage and refrigeration storage.”.

Comment 3: Line 5. Please consider using the plural of the word "quality" here.

Response: Thank you for your comments and suggestions. We have revised our manuscript according to your constructive comments.

In the line 5 of the abstract, "Frozen storage has a long shelf life, but it has a great impact on the quality of meat structure and other quality, and cannot achieve a complete "fresh-keeping" effect." was revised as“Frozen storage has a long shelf life, but it has a great impact on the quality of meat structure and other qualities, and cannot achieve a complete "fresh-keeping" effect. ”.

Comment 4: Lines 7-8. Please consider rewriting this sentence "With the progress of food processing, refrigeration storage and freezing technology, the cold preservation technology between traditional refrigeration storage and frozen storage have attracted more attention, which is ice temperature storage and micro-freezing storage." for a better understanding.

Response: Thank you for your comments. We have revised our manuscript according to your constructive comments.

In the line 7-8 of the abstract, "With the progress of food processing, refrigeration storage and freezing technology, the cold preservation technology between traditional refrigeration storage and frozen storage have attracted more attention, which is ice temperature storage and micro-freezing storage." was revised as“With the development of food processing storage and freezing technology, two new storage methods, ice temperature storage and micro-freezing storage, have attracted more attention.”.

Comment 5: The abstract lacks results (please summarize the article's main findings) and conclusions (please indicate the main conclusions or interpretations).

Response: Thank you for your comments and suggestions. We have revised our manuscript according to your constructive comments.

It is added at end of the abstract,“Finally, this study concluded that the longest shelf life could be achieved by frozen storage, and the best preservation effect was achieved during the shelf life of ice temperature storage, and the effect of micro-freezing storage on the myofibrillar protein oxidation and microstructure was the best.”

Introduction.

Comment 6: Line 7. Is it low cost?

Response: Thank you for your comments and suggestions. Refrigeration storage and  frozen storage can usually be done in a refrigerator or freezer, or even in a naturally low temperature environment, whereas other novel storage methods, such as the ice temperature and micro-freezing methods used in this article, require other expensive equipment.

Comment 7:  Please delete references from the end of the following since these are the results of this study "In this study, fresh beef was selected as the research object. The changes of physicochemical properties, sensory indexes, myofibrillar protein oxidation and microstructure of fresh beef during different low-temperature storage method were studied, and the effects of different low-temperature storage method on the quality of fresh beef were investigated. This work compares and analyzes the advantages and disadvantages of the traditional refrigeration storage and frozen storage with modern ice temperature storage and micro-freezing storage, reveals the mechanism and key points of micro-freezing storage and ice temperature storage, and provides theoretical support for developing new fresh meat preservation technology."

Response: Thank you for your comments and suggestions. At the end of the introduction, we have deleted these references in the revised manuscript according to your constructive comments.

Round 2

Reviewer 1 Report

Thank you for atking up the comments of the reviewers.

Reviewer 2 Report

Thank you for the extensive modifications, you have greatly improved the quality of this paper